# A Volatility Estimator of Stock Market Indices Based on the Intrinsic Entropy Model

**DOI:** 10.3390/e23040484

**Published:** 2021-04-19

**Authors:** Claudiu Vințe, Marcel Ausloos, Titus Felix Furtună

**Affiliations:** 1Department of Economic Informatics and Cybernetics, Bucharest University of Economic Studies, 010552 Bucharest, Romania; felix.furtuna@ie.ase.ro; 2School of Business, Brookfield, University of Leicester, Leicester LE2 1RQ, UK or or marcel.ausloos@uliege.be; 3Department of Statistics and Econometrics, Bucharest University of Economic Studies, 010374 Bucharest, Romania; 4GRAPES, 483 Rue de la Belle Jardiniere, B-4031 Liege, Belgium

**Keywords:** intrinsic entropy model, historical volatility, volatility estimators

## Abstract

Grasping the historical volatility of stock market indices and accurately estimating are two of the major focuses of those involved in the financial securities industry and derivative instruments pricing. This paper presents the results of employing the intrinsic entropy model as a substitute for estimating the volatility of stock market indices. Diverging from the widely used volatility models that take into account only the elements related to the traded prices, namely the open, high, low, and close prices of a trading day (OHLC), the intrinsic entropy model takes into account the traded volumes during the considered time frame as well. We adjust the intraday intrinsic entropy model that we introduced earlier for exchange-traded securities in order to connect daily OHLC prices with the ratio of the corresponding daily volume to the overall volume traded in the considered period. The intrinsic entropy model conceptualizes this ratio as entropic probability or market credence assigned to the corresponding price level. The intrinsic entropy is computed using historical daily data for traded market indices (S&P 500, Dow 30, NYSE Composite, NASDAQ Composite, Nikkei 225, and Hang Seng Index). We compare the results produced by the intrinsic entropy model with the volatility estimates obtained for the same data sets using widely employed industry volatility estimators. The intrinsic entropy model proves to consistently deliver reliable estimates for various time frames while showing peculiarly high values for the coefficient of variation, with the estimates falling in a significantly lower interval range compared with those provided by the other advanced volatility estimators.

## 1. Introduction

When studying the financial securities market, what is of interest for practitioners and investors alike is how much the price of a given instrument varies within a certain time interval. In other words, the dispersion of the price values across the time frame taken into account provides multiple types of information, and is perceived in various ways:(a)Amplitude, between the lowest and highest values in the interval;(b)Deviation from a reference level, being the average price value for the interval for instance;(c)The degree of interest that the instrument receives from the investors, when connected with the traded volume at a given price level;(d)The amplitude of the price changes, in connection with the frequency of changes; frequent movement or slow directional changes following a certain trend, in terms of going up or going down.

These are only a few aspects that naturally derive from following the price variation of a financial instrument in a given time window. This price dispersion over time is identified as the historical volatility of a financial security over a period of time. It has a salient importance in practice for assessing portfolio risk and pricing derivative products [1]. Many different methods have been developed to estimate the historical volatility. These methods use some or all of the usually available daily prices that characterize a traded security: open (O), high (H), low (L), and close (C).

The most common method used to estimate the historical volatility is the close-to-close method. In this approach, the historical volatility is defined as either the annualized variance or standard deviation of log returns [2,3]. In order to keep the presentation consistent with the concept of dispersion, which we employ throughout this paper, the standard deviation of log returns will be compared with benchmark estimators for volatility. If we consider the log return of a traded stock, then:(1)$$x_{i}=ln\left(\frac{c_{i}+d_{i}}{c_{i-1}}\right)$$
where $d_{i}$ is the dividend, which is not adjusted; $c_{i}$ is the closing price of the current time frame (day for instance); and $c_{i-1}$ is the closing price of the previous time frame.

With these assumptions, the classical volatility estimator based on close-to-close prices of *n*-period historical data is given by the standard deviation:(2)$$\sigma =\sqrt{\frac{1}{n}\;\overset{n}{\underset{i=1}{\sum }}{\left(x_{i}-\bar{x}\right)}^{2}}$$
where $\bar{x}=\mu$, the drift, is the average of log returns $x_{i}$ in the period.

Based on close-to-close approach, the trading interval *T* is considered as being the time frame between two consecutive closing prices: from the previous day closing price until the current day closing price. Since within this interval *T* there is an “overnight” period of time during which the market is closed, regardless of the meridian on which the particular stock exchange is located, this duration is commonly modeled as a fraction *f* of the trading interval *T*. Hence, there is an interval of length fT between the previous day’s closing and the current day’s opening, and an interval of length (1−f)T between the current opening and the current closing, during which the market is open for trading.

In terms of notations, we follow the seminal study by Yang and Zhang (2000) [4], as they built their results on the work of Garman and Klass (1980) [5]. As Yang and Zhang mention explicitly in their paper [4], the time fraction *f* does not necessarily quantify the time length of the market closing period, but the fraction *f* is rather meant to model the relative size of the opening jump in comparison to the price evolution during the period of continuous trading. We note that the opening jump may occur due to inclusion of the dividend value when it comes to a traded stock, as in (1) when computing the log returns. Consequently, the notations that we adopt in this paper follow those that were initially used by Garman and Klass [5]:$C_{0}$ or $C_{i-1}$—closing price of the previous day;$O_{1}$ or $O_{i}$—opening price of the current trading day;$H_{1}$ or $H_{i}$—current day’s high, during the trading interval [*f, 1*];$L_{1}$ or $H_{i}$—current day’s low, during the trading interval [*f, 1*];$C_{1}$ or $C_{i}$—closing price of the current day;$o=ln\;O_{1}-ln\;C_{0}$—the normalized opening price;$u=ln\;H_{1}-ln\;O_{1}$—the normalized high of the current period;$d=ln\;L_{1}-ln\;O_{1}$—the normalized low of the current period;$c=ln\;C_{1}-ln\;O_{1}$—the normalized closing price of the current period.

In addition to the above notations, we introduce the following extensions concerning a succession of equally sized *n*-periods *T*:(3)$$o_{i}=ln\left(O_{i}\right)-ln\left(C_{i-1}\right)=ln\left(\frac{O_{i}}{C_{i-1}}\right)\;,\;\;i=\bar{1,\;n}$$
(4)$$u_{i}=ln\left(H_{i}\right)-ln\left(O_{i}\right)=ln\left(\frac{H_{i}}{O_{i}}\right)\;,\;\;i=\bar{1,\;n}$$
(5)$$u_{i}=ln\left(H_{i}\right)-ln\left(O_{i}\right)=ln\left(\frac{H_{i}}{O_{i}}\right)\;,\;\;i=\bar{1,\;n}$$
(6)$$c_{i}=ln\left(C_{i}\right)-ln\left(O_{i}\right)=ln\left(\frac{C_{i}}{O_{i}}\right)\;,\;\;i=\bar{1,\;n}$$

With these notations, the drift or the average of log returns for an *n*-period interval *T* is expressed as:(7)$$\mu =\frac{1}{n}\;\overset{n}{\underset{i=1}{\sum }}\left(o_{i}+c_{i}\right)$$

With this notation, the classical close-to-close volatility estimator becomes:(8)$$V^{CC}=\sqrt{\frac{1}{n}\;\overset{n}{\underset{i=1}{\sum }}{\left[\left(o_{i}+c_{i}\right)-\mu \;\right]}^{2}}$$

The classical close-to-close estimator does handle drift (μ may not be necessarily equal to zero) and quantifies potential opening jumps.

In 1980, Parkinson introduced the first advanced volatility estimator [6] based only on high and low prices (HL), which can be daily, weekly, monthly, or other:(9)$$V^{P}=\sqrt{\frac{1}{n}\;\overset{n}{\underset{i=1}{\sum }}\frac{1}{4\;ln\;2}{\left(u_{i}-d_{i}\right)}^{2}}$$

As it does not take into account the opening jumps, the Parkinson volatility estimator tends to underestimate the volatility. On the other hand, since it does not handle drift (μ=0), in a trendy market $V^{P}$ may overestimate the volatility in the pertinent time interval.

In the same year (1980), and in the same journal issue as Parkinson, Garman and Klass [5] proposed their estimator, which is based on all commonly available prices of the current day of trading (OHLC):(10)$$V^{GK}=\sqrt{\frac{1}{n}\;\overset{n}{\underset{i=1}{\sum }}\left[\frac{1}{2}{\left(ln\frac{H_{i}}{L_{i}}\right)}^{2}-\left(2\;ln2-1\right){\left(ln\frac{C_{i}}{O_{i}}\right)}^{2}\right]}$$

The Garman–Klass estimator includes opening and closing prices for the current trading day. From this perspective, the $V^{GK}$ estimator extends and improves the performance offered by the Parkinson estimator. It does not include the overnight jumps though; therefore, it may underestimate the volatility. If the opening price is not available, the estimator may use the closing price for the previous day of trading. In this context, the $V^{GK}$ estimator handles the overnight jumps but does not isolate potential opening jumps.

Both the Parkinson and Garman–Klass advanced volatility estimators assume that there is no drift (μ=0). In reality, securities may have a noticeable trend for periods of time. In order to overcome this deficiency of the previous estimators, Rogers and Satchell proposed in 1991 [7] a volatility estimator that handles non-zero drifts and which takes into account all of the prices that synthetically characterize a day of trading (OHLC). They refined their estimator in 1994, together with Yoon [8]. With the Garman–Klass notation, the Rogers–Satchell volatility estimator has the flowing formula:(11)$$V^{RS}=\sqrt{\frac{1}{n}\overset{n}{\underset{i=1}{\sum }}\left[u_{i}\left(u_{i}-c_{i}\right)+d_{i}\left(d_{i}-c_{i}\right)\right]}$$

The Rogers–Satchell estimator does not handle opening jumps; therefore, it underestimates the volatility. It accurately explains the volatility portion that can be attributed entirely to a trend in the price evolution. Developing (11) based on (4)–(6), we obtain the following form of Rogers–Satchell volatility estimation, which is simply based on the current day open, high, low, and close prices:(12)$$V^{RS}=\sqrt{\frac{1}{n}\overset{n}{\underset{i=1}{\sum }}\left[ln\left(\frac{H_{i}}{O_{i}}\;\right)ln\left(\frac{H_{i}}{C_{i}}\;\right)+ln\left(\frac{L_{i}}{O_{i}}\;\right)ln\left(\frac{L_{i}}{C_{i}}\right)\right]}$$

According to Rogers and Satchell, the Garman–Klass estimator seems to present two major drawbacks: first, the estimator is biased when there is a non-zero drift rate for the stock return in the period, and second, the empirical observations of stock prices are not continuous, as the Brownian motion model approach stipulates [1]. While the first drawback seems to have no effect since the estimator “works just as well for non-zero (drift rate)”, the second has some consequences. Garman and Klass suggest the use of a given set of values to adjust the figures found when historical volatilities are calculated. However, Rogers and Satchell [8] try to embody the frequency of price observations in the model in order to overcome the drawback. They claim that the corrected estimator outperforms the uncorrected one in a study based on simulated data.

Yang and Zhang noted in [4] that $V^{GK}$ and $V^{RS}$ estimators are arithmetic averages of their corresponding single-period (*n* = 1) estimators, whereas the classical $V^{CC}$ estimator is a multiperiod-based one. They argued that an unbiased variance estimator, which would be both drift-independent and able to handle opening jumps, must be based on multiple periods. Yang and Zhang proposed in 2000 [4] a new minimum-variance, unbiased, multiperiod-based variance estimator (*n* > 1):(13)$$V^{YZ}=\sqrt{V^{O}+k\;V^{C}+\left(1-k\right)\;V^{RS}}$$
where $V^{O}$ and $V^{C}$ are:(14)$$V^{O}=\frac{1}{n}\;\overset{n}{\underset{i=1}{\sum }}{\left(o_{i}-\bar{o}\right)}^{2}$$
(15)$$V^{C}=\frac{1}{n}\;\overset{n}{\underset{i=1}{\sum }}{\left(c_{i}-\bar{c}\right)}^{2}$$
and $\bar{o}=\frac{1}{n}\;\overset{n}{\underset{i=1}{\sum }}o_{i}$, $\bar{c}=\frac{1}{n}\;\overset{n}{\underset{i=1}{\sum }}c_{i}$ are corresponding averages of opening and closing prices in the considered multiperiod, respectively. Yang and Zhang chose the constant *k* in order to minimize the variance of the $V^{YZ}$ estimator:(16)$$k=\frac{0.34}{1.34+\frac{n+1}{n-1}}$$

Yang and Zhang commented in [4] that *k* can never reach zero or one, and this fact proves that neither the classical close-to-close estimator $V^{CC}$ nor the Rogers–Satchell estimator $V^{RS}$ alone has the property of minimum variance. The estimator with minimum variance is a linear combination of both $V^{CC}$ and $V^{RS}$ with positive weights [9]. Yang and Zhang noticed that the weight (1−k) applied on $V^{RS}$ is always greater than the weight *k* applied on $V^{CC}$, which reflects the fact that the variance of $V^{RS}$ is smaller than the variance of $V^{CC}$.

## 2. Materials and Methods

Over the past decades, the use of entropy in modeling various economic phenomena, along with the emergence of econophysics as a scientific discipline [10], has resulted in rapid progress being made in economics outside of the mainstream [11]. Information entropy has been used both to assess the price fluctuations of financial instruments in connection with the maximum entropy distribution [12] or for studding the predictability of stock market returns [13,14].

We conceived the intrinsic entropy model initially based on the intraday trading data, namely the execution data generated by the stock exchange matching engine once one buy and one sell orders are put in correspondence [15]. For each exchange-listed security, a trading day consists of a succession of transactions generated by the exchange matching engine when buy and sell orders meet the conditions for being partially or entirely executed [16,17]. Each individual transaction, namely a trade, consists of the following information: the price at which the trade was made, the executed quantity, and the timestamp at which the order matching occurred and the trade was generated.

With this perspective in mind, the total executed (traded) quantity of a given security is not known until the trading day is over. Therefore, the intrinsic entropy value for a given security is determined every time a new trade is made, and all of the ratios are recalculated for all trades that were made during the day up to the latest one considered at time *t*. Let X be a traded symbol on the market. Based on these considerations, the intraday intrinsic entropy model has the following formalization:(17)$$H_{t}^{X}=-\overset{N_{t}}{\underset{k=1}{\sum }}\left(\frac{p_{k}}{p_{ref}}-1\;\right)\frac{q_{k}}{Q_{t}}\;ln\left(\frac{q_{k}}{Q_{t}}\right)$$
where:$H_{t}^{X}$ is the intrinsic entropy computed for symbol *X* at moment *t*;$N_{t}$ is the total number of trades executed for symbol *X* in the current trading session up to moment *t*;*k* is ordinal trade number;$q_{k}$ is trade quantity, i.e., number of shares of trade *k* for symbol *X*;$p_{k}$ is trade price, i.e., the price of trade *k* for symbol *X*;$Q_{t}$ is the total traded quantity, i.e., the number of shares traded during the day for symbol *X* up to moment *t*, $Q_{t}=\overset{t}{\underset{k=1}{\sum }}q_{k}$;*p_ref_* is reference price for symbol *X*, corresponding to the trading data prior to the moment *t*.

The ratios $\frac{q_{k}}{Q_{t}}$ signify the degree of confidence or support that the market provides to the price level at which the trade was made. The price at which the order matching occurs relative to a certain reference price offers an indication of the inclination of the investors towards buying or selling the considered stock.

The intraday intrinsic entropy model proves to gauge the investors’ interest in a given exchange-traded security. Furthermore, the intrinsic entropy provides an indication regarding the direction and intensity of this interest, either in buying or selling the security. Regarding the employed reference prices, we conclude in [15] that the price of the preceding transaction in the relative price variation $\left(\frac{p_{k}}{p_{k-1}}-1\right)$ provides anchoring to the entropic probability represented by the fraction $\frac{q_{k}}{Q_{t}}$, along with an indication regarding the trading attractiveness in the given security up to the point in time when the intrinsic entropy is computed.

In the case of the intraday trading, the total traded quantity of the entire day is not an a priori known value, hence the intrinsic entropy model proposed for the intraday trading employs ratios that have a moving base. The denominator of the series increases with each exchange-executed quantity for the underlying security:(18)$$\frac{q_{1}}{q_{1}}\;,\;\frac{q_{2}}{q_{1}+q_{2}}\;,\;\frac{q_{3}}{q_{1}+q_{2}+q_{3}}\;,\;\cdots \;,\;\frac{q_{t}}{q_{1}+q_{2}+q_{3}+\cdots +\;q_{t-1}+q_{t}}$$
(19)$$\frac{q_{k}}{Q_{t}}\;,\;\;\;\;Q_{t}=\overset{N_{t}}{\underset{k=1}{\sum }}q_{k}\;,\;\;\;\mathrm{where}\;t\;\mathrm{is}\;a\;\mathrm{timestamp},\;\mathrm{hence}\;\overset{N_{t}}{\underset{k=1}{\sum }}\frac{q_{k}}{Q_{t}}>1$$

Consequently, the value of the fraction becomes smaller as we advance in the trading day and with the number of executed trades, regardless of how big the executed quantity is at any given moment *t*. We note that *t* is a timestamp and does not have the meaning of equally distanced time intervals, since it represents the actual moment during the trading day when the trade was made. Similarly, if we compute the $\frac{q_{k}}{Q_{t}}$ ratios starting with the most recent transaction and go backwards to the very first one from the beginning of the trading day, we obtain the following series:(20)$$\frac{q_{t}}{q_{t}}\;,\;\frac{q_{t-1}}{q_{t}+q_{t-1}}\;,\;\frac{q_{t-2}}{q_{t}+q_{t-1}+q_{t-2}}\;,\;\cdots \;,\;\frac{q_{1}}{q_{t}+q_{t-1}+q_{t-2}+\cdots +\;q_{2}+q_{1}}$$
which produces the same value for the limit:(21)$$\underset{k,\;\;t\to \infty }{\lim}\left(\frac{q_{k}}{Q_{t}}\right)=0\;\;,\;\;\underset{k,\;t\to \infty }{\lim}\left(\frac{q_{k}}{Q_{t}}\ln\left(\frac{q_{k}}{Q_{t}}\right)\right)=0$$

In other words, depending on the direction in which the trading data for the considered period are taken into account, either from the oldest to the most recent or from the most recent to the oldest, the information provided by the traded volume correspondingly favors the older prices or the more recent ones.

Employing the price of the preceding transaction as a reference price preserves the atomicity of each trade within the overall pool of transactions that constitute the trading day on the stock exchange. Consequently, a Markov chain is thereby constructed, in which the price of each individual trade is compared only to the price of the preceding trade.

We proposed a new unbiased volatility estimator based on multiperiod data (*n* > 1) and on the intrinsic entropy model that we introduced in [15].

Developing on this principle, we first considered the intrinsic entropy-based volatility estimator as taking into account only the closing price of the current trading day versus the closing price of the previous day:(22)$$H^{CC}=-\overset{n}{\underset{i=1}{\sum }}\left(\frac{c_{i}}{c_{i-1}}-1\;\right)\;p_{i}\;ln\;p_{i}$$
(23)$$p_{i}=\frac{q_{i}}{Q}\;,\;\;Q=\overset{n}{\underset{i=1}{\sum }}q_{i}\;,\;\;\;\;i=\bar{1,\;n}\;\;\;,\;\;\overset{n}{\underset{i=1}{\sum }}p_{i}=1$$

Based on the meaning of intrinsic entropy introduced in [15] for intraday trading, these daily ratios $\frac{q_{i}}{Q}$ represent the degree of credence that the investors and the market provide to the price levels or to the intensity of price changes. Formula (22) represents an intrinsic entropy-based volatility estimator, which emulates the classical close-to-close approach to volatility.

Diverging from the previously presented volatility models that take into account only the elements related to the traded prices—namely open, high, low, and close prices for a trading day (OHLC)—the intrinsic entropy model takes into account the traded volumes during the considered time frame as well.

We adjusted the intraday intrinsic entropy model that we introduced earlier for exchange-traded securities in order to connect daily OHLC prices with the ratio of the corresponding daily volume to the overall volume traded in the considered period. The intrinsic entropy model conceptualizes this ratio as entropic probability or market credence assigned to the corresponding price level.

The intrinsic entropy is computed using historical daily data for traded market indices (S&P 500, Dow 30, NYSE Composite, NASDAQ Composite, Russell 2000, Hang Seng Index, and Nikkei 225). We compared the results produced by the intrinsic entropy model with the volatility obtained for the same data sets using widely employed industry volatility estimators, namely close-to-close (C), Parkinson (HL), Garman–Klass (OHLC), Rogers–Satchell (OHLC), and Yang–Zhang (OHLC) estimators [18,19].

We consequently studied the efficiency of the intrinsic entropy-based volatility and the other variance-based volatility estimates by comparing them with the volatility of the standard close-to-close estimate. It will be shown that this intrinsic entropy-based volatility model proved to consistently deliver a minimal estimation error, i.e., the minimal variance of the estimates for time frames of 5 to 11 days.

The intrinsic entropy-based model for estimating volatility follows the Yang and Zhang approach regarding the treatment of the overnight jumps, opening jumps, and the drift manifested during the trading day:(24)$$H=\left|\;H^{CO}+k\;H^{OC}+\left(1-k\right)\;H^{OHLC}\;\right|$$
where $H^{CO}$, $H^{OC}$, and $H^{OHLC}$ are the corresponding intrinsic entropies for the overnight, opening, and daily trading hours of the interval *T,* respectively. The intrinsic entropy-based volatility model uses the constant *k* determined by Yang and Zhang with the same purpose of weighting the component that handles the opening jumps $k\;H^{OC}$ and the component that handles the drift $\left(1-k\right)\;H^{OHLC}$. The sum of entropic components may have negative values, and in order to keep the estimates within a comparable spectrum with estimates provided by the other volatility estimators, we take the absolute value of the intrinsic entropy, hence the notation with vertical bars | … |
in Formula (24):(25)$$H^{CO}=-\overset{n}{\underset{i=1}{\sum }}\left(\frac{O_{i}}{C_{i-1}}-1\right)\;p_{i-1}\;ln\;p_{i-1}$$
(26)$$H^{OC}=-\overset{n}{\underset{i=1}{\sum }}\left(\frac{C_{i}}{O_{i}}-1\right)\;p_{i}\;ln\;p_{i}$$
(27)$$H^{OHLC}=-\overset{n}{\underset{i=1}{\sum }}\left[\left(\frac{H_{i}}{O_{i}}-1\right)\left(\frac{H_{i}}{C_{i}}-1\right)+\left(\frac{L_{i}}{O_{i}}-1\right)\left(\frac{L_{i}}{C_{i}}-1\right)\right]\;p_{i}\;ln\;p_{i}$$
where $p_{i}=\frac{q_{i}}{Q}$ are provided by the relations in (23).

The intrinsic entropy-based estimation does not make use of the average of the log returns for an *n*-period interval *T*. Therefore, the estimator that we introduced is independent of drift (μ), and quantifies the overnight and opening jumps. We note that the fractions $p_{i}=\frac{q_{i}}{Q}$, $i=\bar{1,\;n}$ (23) represent the ratio between the daily traded volume $q_{i}$ and the overall traded volume Q of the financial instrument in the considered period. Using log returns provides empirically lower values for the intrinsic entropy-based estimates:(28)$$H^{CO}=-\overset{n}{\underset{i=1}{\sum }}ln\left(\frac{O_{i}}{C_{i-1}}\right)\;p_{i-1}\;ln\;p_{i-1}$$
(29)$$H^{OC}=-\overset{n}{\underset{i=1}{\sum }}ln\left(\frac{C_{i}}{O_{i}}\right)\;p_{i}\;ln\;p_{i}$$
(30)$$H^{OHLC}=-\overset{n}{\underset{i=1}{\sum }}\left[ln\left(\frac{H_{i}}{O_{i}}\;\right)ln\left(\frac{H_{i}}{C_{i}}\;\right)+ln\left(\frac{L_{i}}{O_{i}}\;\right)ln\left(\frac{L_{i}}{C_{i}}\right)\right]\;p_{i}\;ln\;p_{i}$$

In the case of the historical volatility, the *n*-period interval for which it is computed is known a priori, along with daily trading data for the interval. If the intention is to give more weight to the more recent data (prices) then the volume ratios can be computed using (23). This seems to be a more natural approach from the whole market perspective, given that an *n*-period interval is not large enough to completely cancel the contribution of the order data. On the other hand, knowing exactly the number of trading days for which one computes the volatility estimator, a simpler and fairer approach may be more appropriate, for example by calculating the overall traded quantity (volume) from the very beginning, and thereafter performing calculations for the fraction of each day’s volume in the total trade volume of the period. In the intrinsic entropy model, the $p_{i}=\frac{q_{i}}{Q}$ ratios effectively substitute the probabilities in the Shannon’s information entropy formula. Hence, the series of ratios have the following format:(31)$$p_{1}=\frac{q_{1}}{Q}\;,p_{2}=\frac{q_{2}}{Q}\;,\;p_{3}=\frac{q_{3}}{Q}\;,\;\cdots \;,\;p_{n}=\frac{q_{n}}{Q}\;\;\;,\;\overset{n}{\underset{i=1}{\sum }}p_{i}=1,\;\;\;Q=\overset{n}{\underset{i=1}{\sum }}q_{i}$$

We used the probabilities provided by (23) and (31) throughout this paper to compute the intrinsic entropy-based volatility estimator.

Given the fact that the intrinsic entropy-based estimator of historical volatility does not produce results in a comparable range of values with the variance-based estimators, this raises the question regarding how these estimators could actually be compared in a relevant manner that would allow decisive discrimination [20,21].

We note that the intrinsic entropy-based estimates are consistently in a lower range of values compared to the estimates produced by the other volatility estimators, while changing relevantly from one day to another.

The information that is brought in by the daily traded volume and the entropic mechanism through which the intrinsic entropy-based estimations are computed provide for more dynamic changes, although we note that these estimates can offer a more valuable perspective of the overall market evolution for short time horizons. Moreover, the traded volume can be taken into account by investors when focusing on a technical analysis approach [22,23,24]. Figure 1 shows the volatility estimates generated by the intrinsic entropy-based estimator for a 20 day time window of historical data for the S&P 500 market index, along with the index price evolution and the daily traded volume. In comparison, the Yang–Zhang volatility estimator (Figure 2), for the same 20 day time window, provides higher estimates and shows little changes of volatility on a daily basis. If we move to a 60 day time interval, the same pattern is preserved—the intrinsic entropy-based estimator generates volatility in a lower range of values than those produced by the Yang–Zhang estimator, while showing consistent changes in volatility estimates on a daily basis (Figure 3 and Figure 4).

We note that the other volatility estimators that we considered in our analysis, namely the classical close-to-close, Parkinson, Garman–Klass, and Rogers–Satchell estimators, exhibited the same pattern as the Yang–Zhang estimator in terms of providing higher estimates than the intrinsic entropy-based one, and all showed little change of volatility on a daily basis. Furthermore, this pattern is reflected in the volatility estimates computed for all the stock market indices that we took in account (S&P 500, Dow 30, NYSE Composite, NASDAQ Composite, Russell 2000, Hang Seng Index, and Nikkei 225) and for all the *n* day intervals that we considered (5, 10, 15, 20, 30, 60, 90, 150, 260, and 520).

## 3. Results

We now present our empirical findings and compare the estimates produced by the intrinsic entropy-based volatility model against the volatility provided by the classical close-to-close, Parkinson, Garman–Klass, Rogers–Satchell, and Yang–Zhang estimators. We considered for this comparison the historical daily trading data for S&P 500, Dow 30, NYSE Composite, NASDAQ Composite, Russell 2000, Hang Seng Index, and Nikkei 225 indices. The estimates are computed for the following *n*-period intervals, going back from 31 January 2021: 5, 10, 15, 20, 30, 60, 90, 150, 260, and 520. The estimates are computed on a daily basis, by rolling back *n*-period time windows, corresponding to the considered intervals.

As the intrinsic entropy-based model of volatility consistently delivers lower estimates for each time interval and stock market index, we first investigated the following set of indicators to serve for the purpose of comparison, namely the average (Mean), variance (Var), and coefficient of variation (CV).
(32)$$Var=\sigma_{\hat{V}}^{2}=\frac{1}{n}\;\overset{n}{\underset{i=1}{\sum }}{\left({\hat{V}}_{i}-\bar{\hat{V}}\right)}^{2}\;,\;\;\;Mean=\bar{\hat{V}}=\frac{1}{n}\;\overset{n}{\underset{i=1}{\sum }}{\hat{V}}_{i},\;\;\;CV=\frac{\sqrt{Var}}{Mean}$$

These indicators are computed for each volatility estimator ${\hat{V}}_{i}$, stock market index, and time interval. The results are presented in Table 1 for the historical trading data for the S&P 500 index. Along with the shorter time interval, we chose a 260-period time interval in to order encompass an entire trading year and a 520-period time interval for approximating two years of trading data. We also want to investigate the manner in which the volatility estimators reflect the market crash caused by COVID-19 pandemic in the spring of 2020 [25].

Figure 5, Figure 6, Figure 7, Figure 8, Figure 9 and Figure 10 offer a visual perspective of the data contained in Table 1, for time intervals of 5, 10, 15, 20, 30, and 60 days, respectively. We note that the volatility estimates provided by the intrinsic entropy consistently show the mean in a lower range of values, while the coefficient of variation (CV) confirms the earlier observation that the intrinsic entropy estimates change on a daily basis. This peculiar characteristic of the volatility estimates produced by the intrinsic entropy estimator suggests that it may be more useful in estimating the market volatility for short-term trading purposes rather than characterizing the evolution of the historical volatility over the long term.

In Appendix A, Table A1 contains the values of the *Mean*, *Var*, and *CV* of volatility estimates computed for all of the volatility estimators considered for the following stock market indices: Dow 30, NYSE Composite, NASDAQ Composite, Russell 2000, Nikkei 225, and Hang Seng Index. We emphasize the fact that the volatility estimates provided by the intrinsic entropy consistently show the mean in a lower value range, while the coefficient of variation (CV) confirms the earlier observation that the intrinsic entropy estimates change on a daily basis. These empirical results were replicated for all of the stock market indices observed and all of the time intervals considered.

## 4. Discussion

The empirical evidence shows that the volatility estimates based on intrinsic entropy fall in lower value ranges for all of the stock market indices and the time intervals considered; a comparison established on a referential indicator might be worth investigating. In particular, we highlight Molnar, who mentioned in [26] the mean squared error (MSE) and proportional bias (PB). Arnerić et al. [27] employed the MSE as well in their analysis, in order to rank the volatility estimators. For a *n*-period time interval, these functions have the following representations, where $V_{i}$ is the true, unobserved volatility, employed as a benchmark, and ${\hat{V}}_{i}$ is the estimated volatility provided by one of the estimators for each period *i* in the interval:(33)$$MSE=\frac{1}{n}\overset{n}{\underset{i=1}{\sum }}{\left(V_{i}-{\hat{V}}_{i}\right)}^{2}$$
(34)$$PB=\frac{1}{n}\overset{n}{\underset{i=1}{\sum }}\frac{\left|V_{i}-{\hat{V}}_{i}\right|}{V_{i}}$$

We note that not having access to the true, unobserved volatility of the market $V_{i}$, we substituted it with $V_{i}^{CC}$, the classical close-to-close volatility estimator, as a benchmark.

In addition to the MSE and PB indicators, we note the volatility estimators’ efficiency. The efficiency of an estimator is defined as the variance of a benchmark estimator divided by the variance of that particular estimator:(35)$$\mathrm{Efficiency}\;\left(\mathrm{Estimator}\right)=\frac{Var\left(Benchmark\right)}{Var\left(Estimator\right)}$$

Table 2 presents the mean squared error (MSE), proportional bias (PB), and efficiency values for the Parkinson, Garman–Klass, Rogers–Satchell, Yang–Zhang, and intrinsic entropy volatility estimators relative to the classical close-to-close estimator as a benchmark. The computation process uses the S&P 500 stock market index daily trading data for various moving time windows.

We note that the intrinsic entropy-based estimator’s efficiency is consistently higher than the other volatility estimators, particularly for short time intervals of between 5 and 11 days, representing roughly one to two weeks of trading. In order to explore this observation in more detail, we computed the volatility estimators’ efficiency for a series of successive time intervals from 5 to 20 days, along with a 30 day window. Figure 11 shows the evolution of the volatility estimators’ efficiency for the S&P 500 (GSPC) index over these time intervals.

We note that the volatility estimators’ efficiency was not consistent with regard to the stock market indices. The empirical data show no volatility estimator as having the best efficiency for all market indices considered in our analysis. For example, Figure 12 shows the evolution of the volatility estimators’ efficiency for the NYSE Composite (NYA) index over the same 30 day time window.

In Table A2 and Appendix B, we provide the MSE, PB, and efficiency values for the Parkinson, Garman–Klass, Rogers–Satchell, Yang–Zhang, and intrinsic entropy volatility estimators relative to the classical close-to-close estimator as a benchmark for all of the considered stock market indices.

In Appendix C we include Figure A1, Figure A2, Figure A3, Figure A4 and Figure A5, showing the volatility estimators’ efficiency for the Dow 30 (DJI), NASDAQ Composite (IXIC), Russell 2000 (RUT), Nikkei 225 (N225), and Hang Seng Index (HIS) over the 30 day time interval.

We cannot precisely pinpoint the unit of measure for the intrinsic entropy-based estimation of volatility. We perceive this aspect as a limitation of the estimator in the sense that it does measure the dispersion of daily price changes with respect to the daily traded volumes, but not as a pure variance-based estimator; its estimates cannot be directly compared to other volatility estimators that we considered in our research. It does offer a higher coefficient of variance for a lower mean of the estimates, which may suggest a better purpose for its usage as an investment decision support tool, rather than a descriptive reporting tool for historical volatility.

Figure 13 shows the comparative evolution of the Yang–Zhang and intrinsic entropy-based volatility estimators for the S&P 500 stock market index over a time window of 260 days. We want to encompass an entire trading year, with data going backwards from 31 January 2021, in order to reflect the market crash caused by the COVID-19 pandemic in the spring of 2020. We note that the other variance-based volatility estimators, namely the Parkinson, Garman–Klass, Rogers–Satchell, and classical close-to-close estimators, exhibit a similar evolution curve as the Yang–Zhang estimator. The manner in which the intrinsic entropy-based estimator reflects the volatility of “local adjustments” is peculiar. Figure 14 depicts a similar evolution of the Yang–Zhang and intrinsic entropy-based volatility estimators for the NYSE Composite stock market index. In Appendix D, we provide Figure A6, Figure A7, Figure A8, Figure A9 and Figure A10, which show the evolution of Yang–Zhang and intrinsic entropy-based estimates for Dow 30, Russel 2000, NASDAQ Composite, Nikkei 225, and Hang Seng stock market indices, respectively, over a time window of 260 days, going backwards from 31 January 2021.

From this different perspective of market volatility, we note that the intrinsic entropy-based volatility estimates may have a more useful role in emphasizing the fractured nature of the market [28,29]. Robert A. Levy argued in [30] that even if we assume the efficient market hypothesis advocated by Eugene Fama [31,32] to be at play, intercorrelations or co-movements in securities prices could conceal existing dependencies in successive price changes. Levy conducted a serial correlation study of securities performance ranks and reached the conclusion that this technique offers a better indication regarding close relationships between certain securities over time than a similar study of successive first differences would provide. Levy concluded in [30] that even if one adheres to the efficient market hypothesis, his findings regarding the superior profits that can be achieved by investing in securities, which historically have had strong price movements, do not necessarily contradict the random walks hypothesis. Perhaps this observation of “relatively strong price movements” has something to do with the interest that the investors show in those securities, something which may be due to the fundamentals of those securities. The intrinsic value of these securities appears to consistently drag investors towards them. It is the market perception that their underlying companies are the leaders of the field they are activating in. Essentially, Levy’s suggestion is an early hint to the fractal theory investigated more recently through econophysics methods [33].

Levy also commented in [30] that the best results were obtained when dealing with the most volatile stocks, an aspect emphasized by Myers as well in [34]. Corroborating Levy’s observations with the characteristics of the estimates produced by the intrinsic entropy-based volatility, we note that the currently identified limitations regarding the precise nature of the estimator’s unit of measure and its high variability within a low mean of the estimates could provide a complementary perspective of the market in comparison with the variance-based volatility estimators. We also notice that the stock market indices containing fewer constituents, namely the Dow Jones Industrial Average (30 listed companies) and Hang Seng Index (50 constituents), exhibited a higher level of uncertainty (Figure A6 and Figure A10 in Appendix D) during the 2020 COVID-19 crisis compared to the considered indices with higher numbers of constituents [35].

## 5. Conclusions

This paper presents the results from employing the intrinsic entropy model for volatility estimation of stock market indices. Diverging from the widely used volatility models that take into account only the elements related to the traded prices, —namely the open, high, low, and close prices of a trading day (OHLC)—the intrinsic entropy model includes the traded volumes during the considered time frame as well. We adjusted the intraday intrinsic entropy model that we introduced earlier for exchange-traded securities in order to connect daily OHLC prices with the ratio of the corresponding daily volume to the overall volume traded in the considered period. The intrinsic entropy model conceptualizes this ratio as an entropic probability or market credence assigned to the corresponding price level. The intrinsic entropy is computed using historical daily data for traded market indices (S&P 500, Dow 30, NYSE Composite, NASDAQ Composite, Russell 2000, Nikkei 225, and Hang Seng Index). We compared the results produced by the intrinsic entropy model with the volatility obtained for the same data sets using widely employed in the markets volatility estimators. The intrinsic entropy model proved to consistently deliver lower volatility estimations for various time frames we experimented with, compared with those provided by the other advanced volatility estimators. We note that while producing estimates in a significantly lower range compared with the other considered volatility estimators, the intrinsic entropy-based volatility offers consistently higher values for the coefficient of variation of its estimates. The tests that we conducted using historical trading data concerning the major international stock market indices provide empirical evidence that the intrinsic entropy-based volatility estimator offers more information regarding the market volatility, particularly for short time intervals of 5 to 11 days of trading data.

We comment that the identified limitations of the intrinsic entropy-based volatility estimator, namely the precise nature of its unit of measure and its high variability within a low average of the estimates, could provide a complementary perspective of the market in comparison with the variance-based volatility estimators.

## Figures and Tables

**Figure 1 entropy-23-00484-f001:**
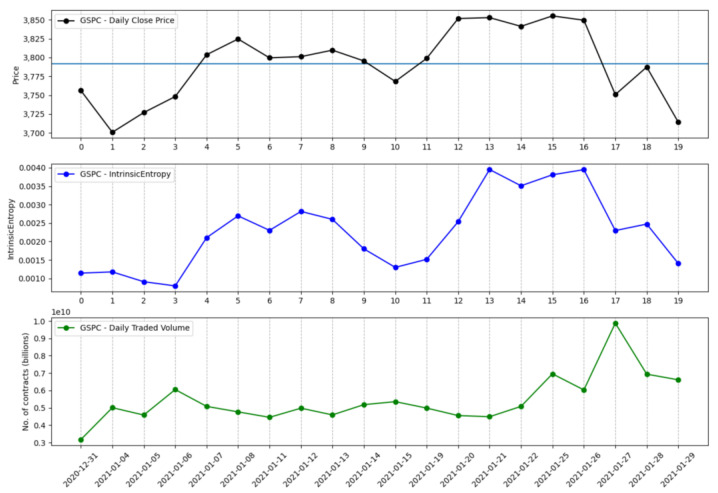
S&P 500 volatility for a 20 day time interval produced by the intrinsic entropy-based estimator.

**Figure 2 entropy-23-00484-f002:**
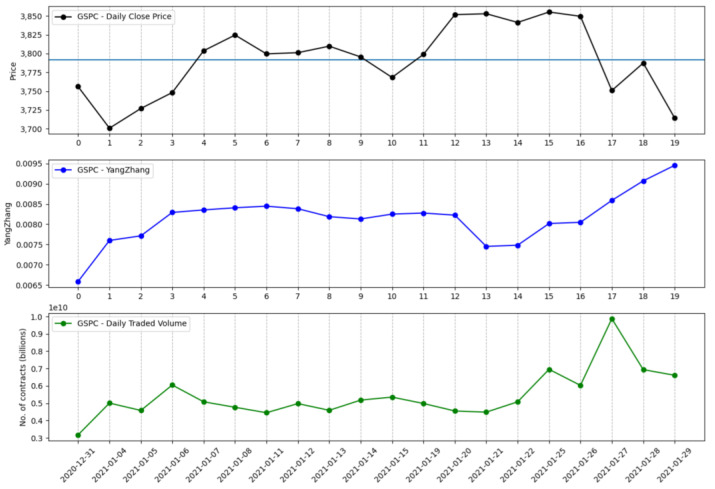
S&P 500 volatility for a 20 day time interval produced by the Yang–Zhang estimator.

**Figure 3 entropy-23-00484-f003:**
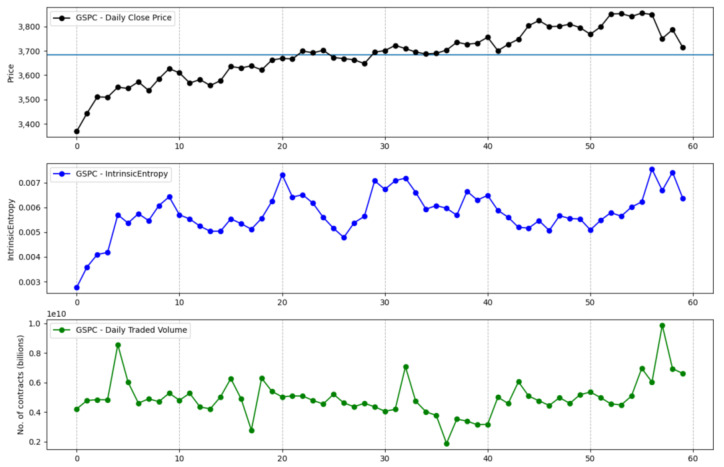
S&P 500 volatility for a 60 day time interval produced by the intrinsic entropy-based estimator.

**Figure 4 entropy-23-00484-f004:**
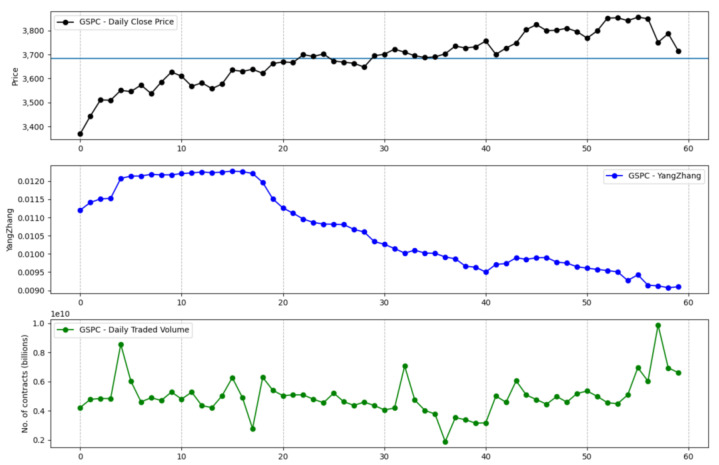
S&P 500 volatility for a 60 day time interval produced by the Yang–Zhang estimator.

**Figure 5 entropy-23-00484-f005:**
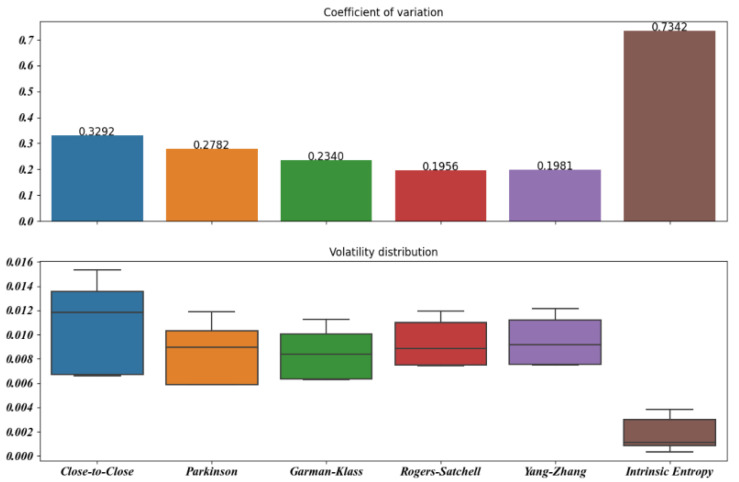
Mean, Var, and CV of S&P 500 volatility estimates for 5 day time windows.

**Figure 6 entropy-23-00484-f006:**
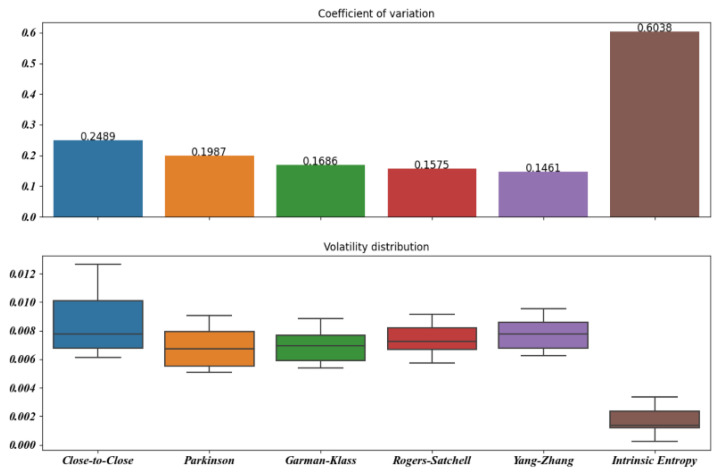
Mean, Var, and CV of S&P 500 volatility estimates for 10 day time windows.

**Figure 7 entropy-23-00484-f007:**
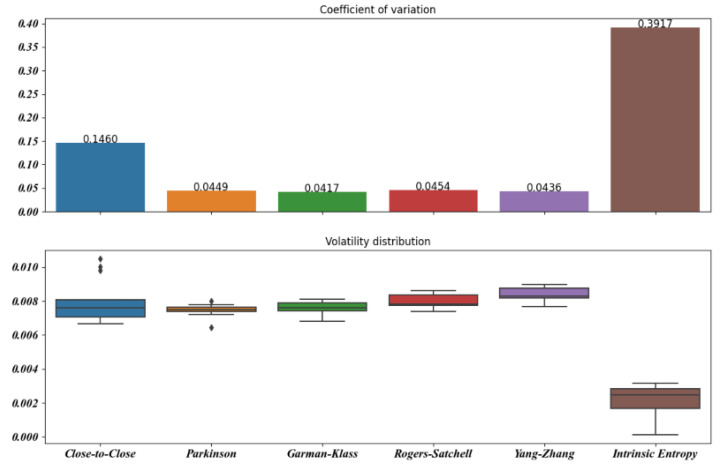
Mean, Var, and CV of S&P 500 volatility estimates for 15 day time windows.

**Figure 8 entropy-23-00484-f008:**
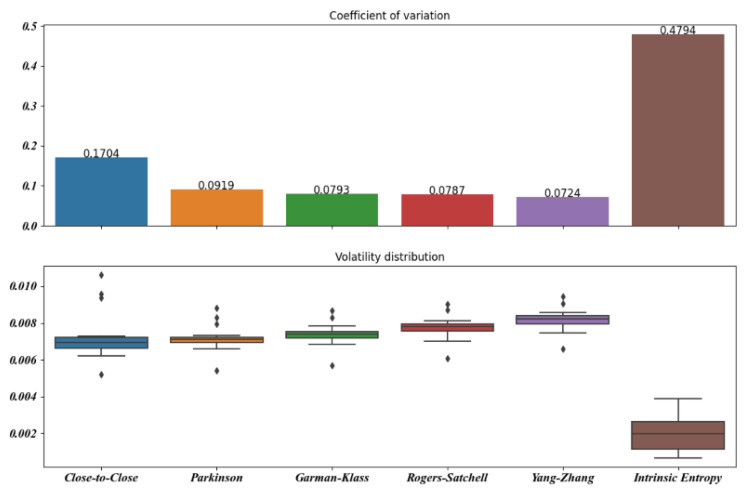
Mean, Var, and CV of S&P 500 volatility estimates for 20 day time windows.

**Figure 9 entropy-23-00484-f009:**
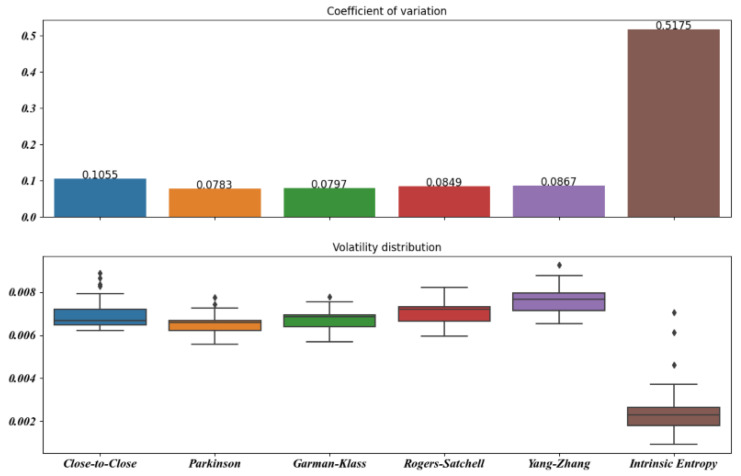
Mean, Var, and CV of S&P 500 volatility estimates for 30 day time windows.

**Figure 10 entropy-23-00484-f010:**
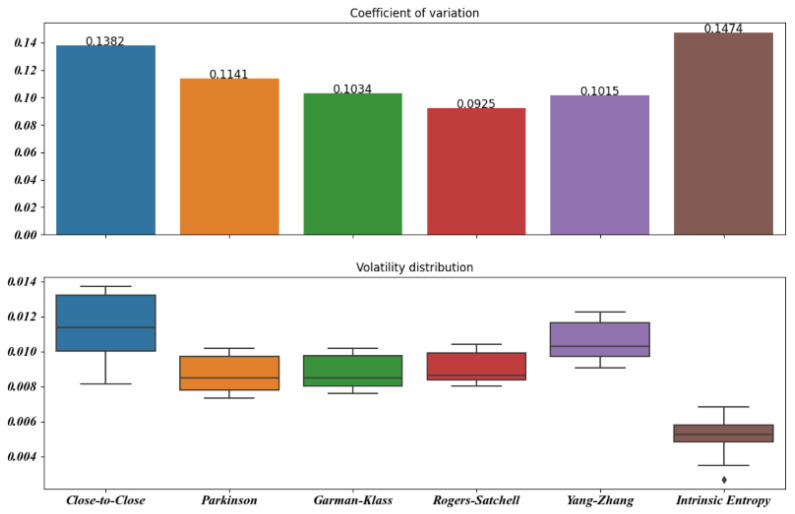
Mean, Var, and CV of S&P 500 volatility estimates for 60 day time windows.

**Figure 11 entropy-23-00484-f011:**
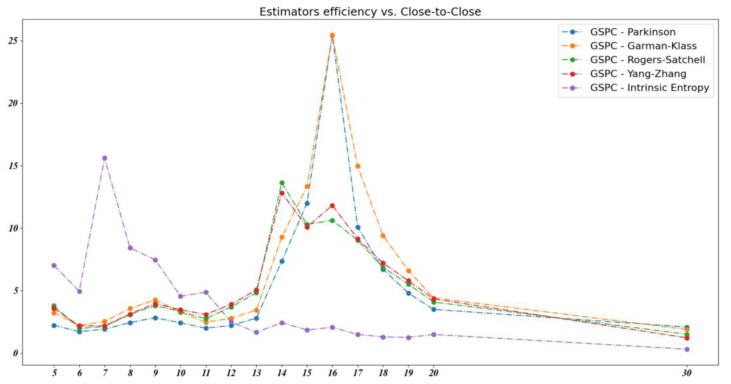
The volatility estimators’ efficiency for the S&P 500 index over a 30 day time window.

**Figure 12 entropy-23-00484-f012:**
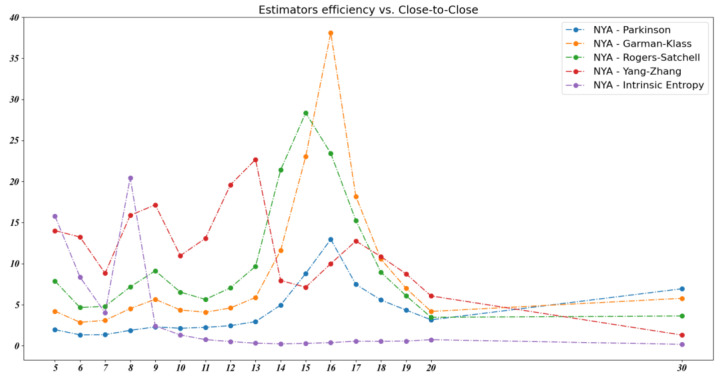
The volatility estimators’ efficiency for the NYSE Composite index over a 30 day time window.

**Figure 13 entropy-23-00484-f013:**
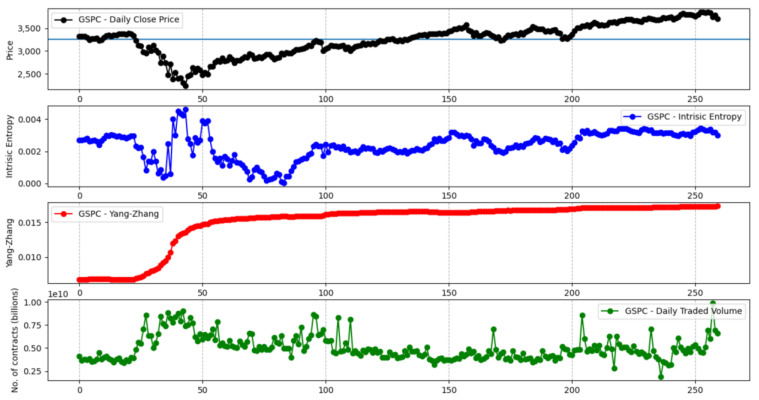
Yang–Zhang and intrinsic entropy-based estimates for the S&P 500 stock market index over a 260 day time window.

**Figure 14 entropy-23-00484-f014:**
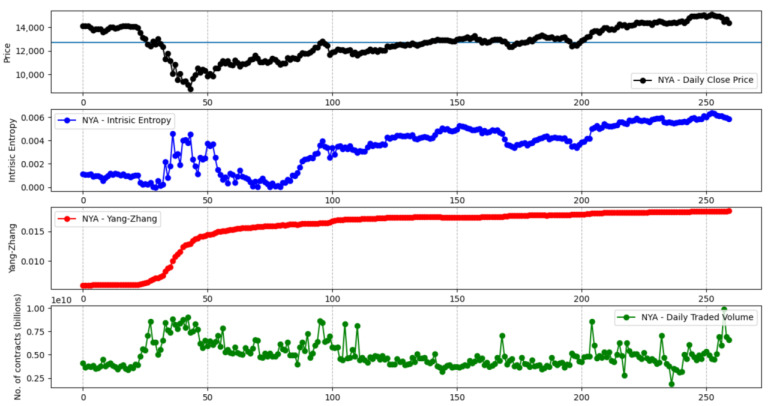
Yang–Zhang and intrinsic entropy-based estimates for the NYSE Composite stock market index over a 260 day time window.

**Table 1 entropy-23-00484-t001:** Comparison of volatility indicators’ main statistical characteristics, namely the mean, variance, and CV, for the S&P 500 stock market index.

n-day period	Indicator	Close-to-Close	Parkinson	Garman–Klass	Rogers–Satchell	Yang–Zhang	Intrinsic Entropy
5	Mean	0.01081191	0.00858391	0.00846942	0.00934932	0.00950730	0.00183014
	Var	0.00001267	0.00000570	0.00000393	0.00000334	0.00000355	0.00000181
	CV	0.32923308	0.27821303	0.23396580	0.19555159	0.19805620	0.73423570
10	Mean	0.00848572	0.00682842	0.00692432	0.00735007	0.00776445	0.00164044
	Var	0.00000446	0.00000184	0.00000136	0.00000134	0.00000129	0.00000098
	CV	0.24886403	0.19870065	0.16862562	0.15752465	0.14611841	0.60381781
15	Mean	0.00796663	0.00747017	0.00762885	0.00797837	0.00839386	0.00218300
	Var	0.00000135	0.00000011	0.00000010	0.00000013	0.00000013	0.00000073
	CV	0.14600436	0.04494159	0.04173677	0.04543190	0.04363811	0.39170477
20	Mean	0.00720326	0.00714908	0.00736426	0.00771872	0.00814834	0.00210023
	Var	0.00000151	0.00000043	0.00000034	0.00000037	0.00000035	0.00000101
	CV	0.17040678	0.09185855	0.07930043	0.07867658	0.07236708	0.47937098
30	Mean	0.00697398	0.00650756	0.00671276	0.00705408	0.00768235	0.00255213
	Var	0.00000054	0.00000026	0.00000029	0.00000036	0.00000044	0.00000174
	CV	0.10551644	0.07828882	0.07965612	0.08491295	0.08667840	0.51750714
60	Mean	0.01139396	0.00866943	0.00879631	0.00909731	0.01064884	0.00532412
	Var	0.00000248	0.00000098	0.00000083	0.00000071	0.00000117	0.00000062
	CV	0.13816747	0.11409147	0.10337760	0.09249231	0.10152583	0.14735764
90	Mean	0.01174813	0.00909485	0.00908652	0.00925370	0.01090736	0.00491880
	Var	0.00000061	0.00000026	0.00000018	0.00000012	0.00000035	0.00000049
	CV	0.06669324	0.05629938	0.04662809	0.03702058	0.05431710	0.14238142
150	Mean	0.02026517	0.01292374	0.01268508	0.01270180	0.01662604	0.00425425
	Var	0.00004805	0.00000982	0.00000895	0.00000848	0.00002077	0.00001062
	CV	0.34207228	0.24244449	0.23587519	0.22924472	0.27411918	0.76600130
260	Mean	0.01900556	0.01154922	0.01130363	0.01129923	0.01504082	0.00216291
	Var	0.00002038	0.00000577	0.00000544	0.00000534	0.00001086	0.00000067
	CV	0.23753649	0.20792360	0.20626915	0.20451721	0.21905171	0.37938593
520	Mean	0.01206458	0.00858504	0.00840142	0.00836915	0.01025407	0.00195492
	Var	0.00001327	0.00000256	0.00000251	0.00000254	0.00000642	0.00000039
	CV	0.30195812	0.18654889	0.18840840	0.19061419	0.24702965	0.31958341

**Table 2 entropy-23-00484-t002:** Comparison of volatility indicators for MSE, PB, and efficiency values for the S&P 500 stock market index, considering the close-to-close estimator as a benchmark.

**n-day period**	Indicator	Parkinson	Garman–Klass	Rogers–Satchell	Yang–Zhang	Intrinsic Entropy
5	MSE	0.00000639	0.00000818	0.00000564	0.00000488	0.00010374
	PB	0.18841732	0.18392193	0.18060307	0.17284327	0.76078733
	Efficiency	2.22170586	3.22700215	3.79078219	3.57371914	7.01730423
10	MSE	0.00000423	0.00000471	0.00000367	0.00000267	0.00005435
	PB	0.18139121	0.16823886	0.15273526	0.13357895	0.78269438
	Efficiency	2.42249618	3.27115119	3.32675961	3.46473785	4.54539052
15	MSE	0.00000132	0.00000151	0.00000136	0.00000147	0.00003661
	PB	0.08677112	0.09596655	0.11311439	0.13535058	0.71360683
	Efficiency	12.00391935	13.34518672	10.29748059	10.08383070	1.85035735
20	MSE	0.00000049	0.00000071	0.00000101	0.00000163	0.00002848
	PB	0.06976201	0.09452474	0.13010625	0.17429379	0.70464018
	Efficiency	3.49376290	4.41797208	4.08555032	4.33322901	1.48647158
30	MSE	0.00000058	0.00000049	0.00000045	0.00000083	0.00002091
	PB	0.08109629	0.08418711	0.08855724	0.11604767	0.63921308
	Efficiency	2.08625170	1.89392060	1.50929350	1.22121439	0.31042934
60	MSE	0.00000782	0.00000727	0.00000592	0.00000086	0.00004082
	PB	0.23606244	0.22358785	0.19592847	0.06647757	0.52001945
	Efficiency	2.53322216	2.99713613	3.50044573	2.12033020	4.02642459
90	MSE	0.00000718	0.00000727	0.00000648	0.00000080	0.00004764
	PB	0.22508669	0.22533215	0.21056193	0.07097473	0.57981658
	Efficiency	2.34154450	3.41988203	5.23098789	1.74899576	1.25163039
150	MSE	0.00006840	0.00007309	0.00007350	0.00001893	0.00035241
	PB	0.33503046	0.34542667	0.34263106	0.15609292	0.70799995
	Efficiency	4.89480315	5.36766037	5.66770065	2.31355323	4.52512182
260	MSE	0.00006018	0.00006422	0.00006439	0.00001729	0.00030523
	PB	0.38206067	0.39451774	0.39413663	0.19999741	0.87010186
	Efficiency	3.53434983	3.74902165	3.81649668	1.87752467	30.26783213
520	MSE	0.00001630	0.00001769	0.00001788	0.00000452	0.00011739
	PB	0.26329038	0.27943249	0.28265855	0.13576132	0.81734381
	Efficiency	5.17426017	5.29677442	5.21488373	2.06836292	34.00095019

## Data Availability

Data is available from the corresponding author upon reasonable request.

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
