# Peer review of "A Volatility Estimator of Stock Market Indices Based on the Intrinsic Entropy Model"

_entropy, 2021, doi:10.3390/e23040484_

Round 1

Reviewer 1 Report

The article focuses on the problem of how to capture more consistently the historical volatility of stock market indices, and also how to estimate it with greater precision. In this sense, the article presents very clearly, and with an interesting  step by step approach to the problem and the background,  the main existing references on the historical volatility of stock market indices, and a novel method is proposed that offers a estimate of lower volatilities for various time frames, based on the use of an intraday intrinsic entropy model, which would offer an alternative for estimating volatility, which would also have the advantage of including the traded volumes during the considered time frame, while conventional indices only consider elements related to the traded prices (namely Open, High, Low, Close prices of a trading day -OHLC).

The authors conceived this intrinsic entropy model, as an emerging model adjusted from their work published in 2019 related to exchange-traded securities, and where they incorporate the daily OHLC prices with the ratio of the corresponding daily volume to the overall volume traded in the respective period considered. The intrinsic entropy model has as a virtue that it proposes a ratio, understood as the entropic probability (market credence that is assigned to the corresponding price level).

The authors calculate this entropic probability ratio using historical daily data for traded market indices (S&P 500, Dow 30, NYSE Composite, 22
NASDAQ Composite, Nikkei 225 and Hang Seng Index), and compare it with the results of volatility estimators of conventional use, with the same data set. It should be noted that this intraday intrinsic entropy model consistently provides reliable estimates in various time frames with which the authors experimented, and in particular offers more information regarding the
market volatility, especially for short time intervals, 5 to 11 days of trading data.

The article is very well written, and its line of discourse make it easier for the reader to increase the understanding of the complexity in the analysis, and also to recognize the limitations and advantages of conventional estimators. The treatment of the antecedents, problem statement, methods, discussion and results, I find them rigorous, relevant, pertinent, and with a high scientific and technical level, around a central question for industry and academia. Respectfully, and without this representing a condition for the publication of the text in its current condition, it is my wish to propose to the authors two brief improvements for the enrichment of the document:
- Briefly present the Intrinsic Entropy Model for Exchange-Traded Securities of 2019, as a precursor to the model under discussion. This seems pertinent to me, not only to invite the reader to inquire about the 2019 article, but because the readers of this paper may not necessarily be well versed in the previous publication.
- Briefly establish the limitations that the intrinsic entropy model may eventually have in some analysis contexts.

Reviewer 2 Report

The article is well written and presents topic which is up-to-date and interesting for investors. The main achievements of the authors is supplementing the existiing models by the traded volumes that could also influence the financial market changes. The literature review is sufficient. Hovever, the authors should:

better describe the novelty of the model and results in the context of existing references, 

indicate the limitations of using this model,

present the conditions and possibilities of using the model in decision making processes on financial markets.
